# Diagnostic accuracy of cytology for the detection of endometrial cancer in urine and vaginal samples

Helena O'Flynn[1], Neil A. J. Ryan[1], Nadira Narine [2], David Shelton[2], Durgesh Rana[2] & Emma J. Crosbie [1,3✉]

Postmenopausal bleeding triggers urgent investigation by sequential invasive tests that are avoidable for the 90–95% of women who do not have endometrial cancer. A simple, non-invasive tool that accurately identifies cancer and safely reassures healthy women could transform patient care. Here we report, in a cross-sectional diagnostic accuracy study of 103 women with known cancer and 113 with unexplained postmenopausal bleeding, that urine and vaginal cytology has a combined sensitivity of 91.7% (95% CI 85.0%, 96.1%) and specificity of 88.8% (81.2%, 94.1%) for gynecological cancer detection. Cytology identifies 91 endometrial, two fallopian tube and one cervical cancer from 103 known cancer cases. In women with unexplained postmenopausal bleeding, cytology identifies all four endometrial cancers and three others (cervical, ovarian and bladder), for a 12/107 (11.2%) false positive rate. We show proof-of-principle that endometrial cancer can be detected in urine and vaginal fluid. Prospective validation of these findings will support incorporation of this non-invasive test into clinical practice.

[1] Gynaecological Oncology Research Group, Division of Cancer Sciences, Faculty of Biology, Medicine and Health, University of Manchester, St Mary's Hospital, Manchester, UK. [2] Manchester Cytology Centre, Manchester University NHS Foundation Trust, Manchester, UK. [3] Division of Gynaecology, St Mary's Hospital, Manchester University NHS Foundation Trust, Manchester Academic Health Science Centre, Manchester, UK. ✉email: Emma. crosbie@manchester.ac.uk

Endometrial cancer is the sixth most common cancer affecting women globally, with approximately 382,000 new diagnoses and 89,900 deaths from the disease in 2018 (ref. [1]). Most women present early following the onset of post-menopausal bleeding (PMB) and have an excellent prognosis, but 20% of women present with advanced disease, for whom 5-year survival rates are just 15%[2]. New strategies to facilitate early diagnosis are urgently needed to enable curative hysterectomy for women who present with biologically aggressive disease[3].

Although the cardinal symptom of endometrial cancer, only 5–10% of women with PMB have sinister underlying pathology[4]. Current practice is to exclude malignant disease through sequential transvaginal ultrasound scan (TVS), outpatient hysteroscopy and endometrial biopsy; however, this diagnostic pathway has limitations. TVS lacks specificity as a triage tool, exposing a high proportion of healthy women to further tests[5]. Failed outpatient hysteroscopy and endometrial biopsy, mostly due to technical failure or intolerable pain, affects up to 31% of women, demanding repeat investigations under general anaesthetic[6–8]. Currently millions of women who do not have endometrial cancer undergo these invasive tests every year, with huge financial implications for health service providers[9] and at a considerable personal cost to women[10]. A simple, easy to administer, non-invasive test that can triage women with endometrial cancer for diagnostic testing whilst safely reassuring the vast majority of healthy women, would improve patient care.

Endometrial cancer is known to shed malignant cells through the cervix into the lower genital tract. Tumour DNA collected from cervical brushes and vaginal tampons demonstrate methylation levels and mutational profiles matching those of the endometrial cancer resected at hysterectomy[11–13]. Tumour cells have also been reported on routine cervical cytology samples in 45% of women with endometrial cancer[14] although cytology has not been formally tested in this regard. Cytology could be a useful tool for endometrial cancer detection because the expertise and infrastructure required to deliver it is already established in most healthcare settings. Advantages include low costs, quick turn-around times and the potential for point-of-care diagnosis[15,16]. In Japan, endometrial cytology is a well-established endometrial cancer diagnostic tool, although it relies on invasive intrauterine sampling[17,18].

Here, we test the hypothesis that PMB offers a potential liquid biopsy to enable endometrial cancer detection in lower urogenital tract samples. We provide proof-of-principle that endometrial cancer can be detected by cytology in voided urine and non-invasive vaginal samples with high diagnostic accuracy, when assessors are blinded to cancer outcomes.

## Results

**Study participants**. In total, 216 women participated in the study, including 103 with known or suspected endometrial cancer and 113 with unexplained PMB (Fig. 1). Their median age and BMI was 61 years (IQR 54, 96) and 30 kg/m$^2$ (IQR 25, 36), respectively, and they were of predominantly White British ethnicity (77.3% White, 10.6% Asian, 6.5% Afro-Caribbean, 5.6% other). Women with endometrial cancer were older (mean 68.5 years vs. 58 years, $p < 0.001$) and more obese (mean 33 kg/m$^2$ vs. 30 kg/m$^2$, $p = 0.005$) than their PMB counterparts (Table 1).

**Urine and vaginal sampling**. All 216 women provided at least one urogenital sample. Matched urine and vaginal samples were obtained from 184/216 women (85%). Vaginal samples were not available for 13 women (6.0%) and urine samples were not available for 13 women (6.0%). Six urine samples were considered inadequate for cytology due to paucicellular content on microscopic review.

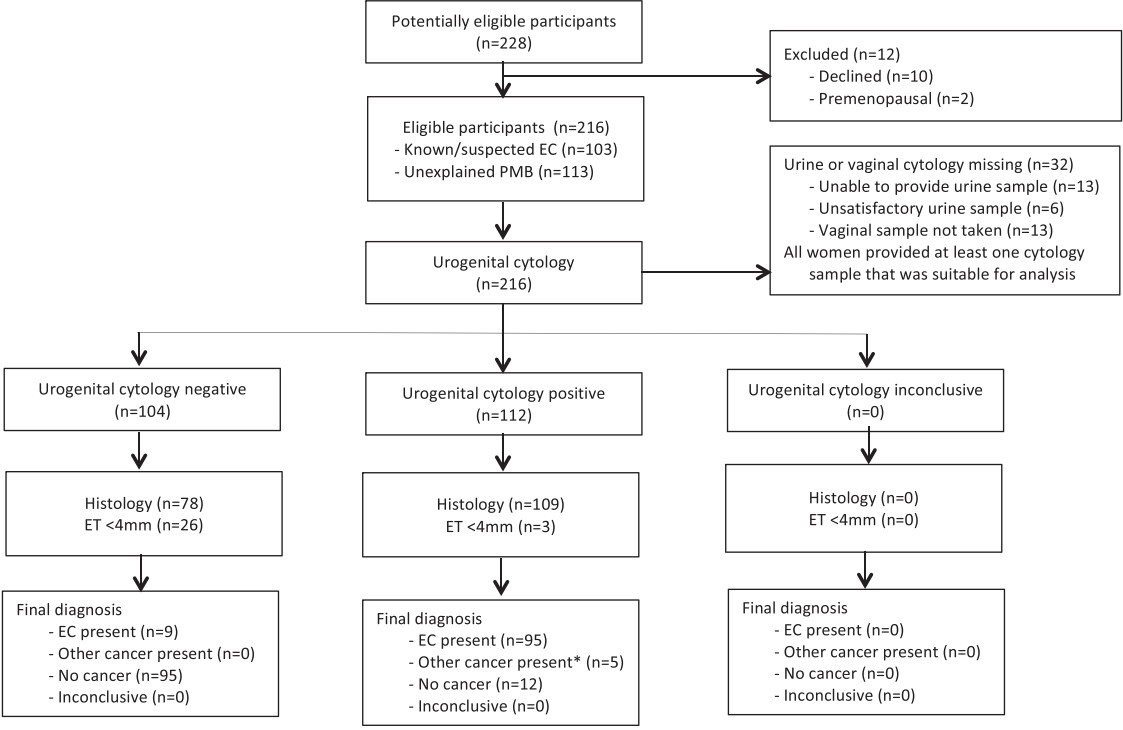

**Fig. 1 STARD diagram showing flow of participants through the study.** Key. ET, endometrial thickness; ET <4 mm, histology not indicated; EC, endometrial cancer; PMB, postmenopausal bleeding. *Other cancers: ovarian ($n = 1$), fallopian tube ($n = 2$), cervical ($n = 2$). Cytology scoring: positive (vaginal and/or urine cytology positive); negative (both vaginal and urine cytology negative); inconclusive (both vaginal and urine cytology inadequate for assessment).

**Table 1 Baseline characteristics of study population.**

| Variable | EC<br>n = 103<br>n (%) | PMB<br>n = 113<br>n (%) | All<br>n = 216<br>n (%) |
|---|---|---|---|
| Age at diagnosis | | | |
| <60 years | 24 (23.3) | 79 (69.9) | 103 (47.7) |
| 60–70 years | 32 (31.1) | 18 (15.9) | 50 (23.1) |
| >70 years | 47 (45.6) | 16 (14.2) | 63 (29.2) |
| BMI at diagnosis | | | |
| <25 kg/m$^2$ | 15 (14.6) | 34 (30.1) | 49 (22.7) |
| 25–29.9 kg/m$^2$ | 25 (24.3) | 29 (25.7) | 54 (25.0) |
| ≥30 kg/m$^2$ | 63 (61.2) | 46 (40.7) | 109 (50.5) |
| Missing data | | 4 (3.5) | 4 (1.9) |
| Presenting complaint | | | |
| PMB | 89 (86.4) | 113 (100) | 202 (93.5) |
| Abnormal bleeding – other[a] | 5 (4.9) | | 5 (2.3) |
| Other[b] | 9 (8.7) | | 9 (4.2) |
| Diagnosis | | | |
| Endometrial cancer | 100 (97.1) | 4 (3.5) | 104 (48.1) |
| Atypical hyperplasia | 1 (1.0) | 3 (2.7) | 3 (1.4) |
| Cervical cancer | 2 (1.9) | 1 (0.9) | 2 (0.9) |
| Ovarian cancer | | 1 (0.9) | 1 (0.5) |
| Fallopian tube cancer | | 0 | 2 (0.9) |
| Benign endometrial polyp | | 31 (27.4) | 31 (14.4) |
| Benign cervical polyp | | 6 (5.3) | 6 (2.9) |
| Vulvovaginal atrophy | | 39 (34.5) | 39 (18.1) |
| Unscheduled bleeding on HRT | | 5 (4.4) | 5 (2.3) |
| Other | | 2 (1.8) | 2 (0.9) |
| No cause for bleeding found | | 21 (18.6) | 21 (9.7) |
| EC Cases | | | |
| Histological subtype | 58 (58.0) | 3 (75.0) | 61 (28.2) |
| Endometrioid | 42 (42.0) | 1 (25.0) | 43 (20.0) |
| Non-endometrioid | 12 (12.0) | 0 | 12 (5.6) |
| Serous | 7 (7.0) | 0 | 7 (2.8) |
| Clear cell | 15 (15.0) | 1 (25.0) | 16 (7.4) |
| Carcinosarcoma | 7 (7.0) | 0 | 7 (3.3) |
| Mixed Mucinous | 1 (1.0) | 0 | 1 (0.5) |
| EC Cases | | | |
| Grade 1 | 36 (36.0) | 2 (50.0) | 38 (17.6) |
| 2 | 17 (17.0) | 1 (25.0) | 18 (8.4) |
| 3 | 47 (47.0) | 1 (25.0) | 48 (22.2) |
| EC Cases | | | |
| FIGO (2009) Stage 1 | 65 (65.0) | 3 (75.0) | 68 (31.5) |
| 2 | 13 (13.0) | 0 | 13 (6.0) |
| 3 | 18 (18.0) | 1 (25.0) | 19 (8.8) |
| 4 | 4 (4.0) | 0 | 4 (1.9) |
| Lymphovascular space invasion | | | |
| Absent | 58 (58.0) | 2 (50.0) | 60 (27.8) |
| Present | 35 (35.0) | 2 (50.0) | 37 (17.1) |
| Missing data[c] | 7 (7.0) | | 7 (3.2) |
| Depth of myometrial invasion | | | |
| <50% | 53 (53.00) | 2 (50.0) | 55 (25.5) |
| ≥50% | 41 (41.00) | 2 (50.0) | 43 (19.9) |
| Missing data[c] | 6 (6.0) | | 6 (2.8) |

*EC* endometrial cancer, *PMB* postmenopausal bleeding, *BMI* body mass index, *HRT* hormone replacement therapy.
[a]Abnormal pre- or peri-menopausal bleeding.
[b]Haematuria, haematocolpos, pelvic mass, abdominal bloating or pain, subfertility.
[c]Including five women who did not have surgery, e.g. for advanced disease, fertility-sparing or medical fitness reasons.

**Clinical diagnosis reference standard.** Ninety-eight out of 103 women with suspected or known endometrial cancer underwent hysterectomy and were staged by the FIGO 2009 classification system. One patient was not fit for surgery and a further 4 grade 1 stage 1a endometrioid endometrial cancer were managed with intrauterine progestin (levonorgestrel-releasing intrauterine system) for fertility-sparing reasons.

Eighty-four out of 113 (74.3%) women with unexplained PMB had an endometrial biopsy; for seven women, this confirmed endometrial cancer (4/113, 3.5%) or atypical hyperplasia (3/113, 2.7%). One woman (0.9%) had a suspicious cervix on speculum examination and a cervical biopsy showed invasive squamous cell carcinoma of the cervix. One woman had a complex solid/cystic pelvi-abdominal mass and large volume ascites on TVS, and subsequently underwent ultrasound-guided omental biopsy followed by neoadjuvant chemotherapy and interval debulking surgery for biopsy-proven high-grade serous adenocarcinoma of the ovary. One woman had a malignant endometrial biopsy and subsequent MRI scan, cystoscopy and transurethral bladder biopsy confirmed the presence of a synchronous 1 cm G3 transitional cell carcinoma (TCC) and carcinoma in situ (CIS). The remaining women with PMB had a negative endometrial biopsy (75, 66.3%) or an endometrial thickness of <4 mm (29, 25.7%), for whom a biopsy was not indicated. No women investigated for PMB and found to have no sinister underlying pathology were subsequently diagnosed with a gynaecological cancer during the study period.

**Urine and vaginal sample analyses.** Microscopic haematuria was more common in women with gynaecological cancer than those without (77% versus 56%, p = 0.04). Both urine and vaginal cytology samples showed mixed cellular populations consisting predominantly of benign urothelial and/or squamous epithelial and inflammatory cells. Malignant endometrial cancer cells appeared as clusters of hyperchromatic glandular cells with large, atypical nuclei (Fig. 2). These cells exhibited an immunoprofile consistent with female genital tract (CK7+, CK20−, PAX-8+, ER+) rather than urinary tract (GATA3−) origin. Samples varied in cellularity and the proportion of malignant endometrial cancer cells seen, with vaginal samples in general showing more frequent and larger clusters of malignant cells than their matched urine sample.

**Diagnostic accuracy of urine and vaginal cytology.** Of the 103 women with suspected endometrial cancer, matched urine and vaginal samples were available for 93 women, vaginal samples only for 8 women and a urine sample only for 2 women. More than half of the endometrial cancer patients had low grade (53%), early stage (65%) disease of endometrioid histological subtype (58%) (Table 1). Urine and/or vaginal cytology was positive in 94 of 103 cases of which two were reported as showing abnormal cells of unknown significance (ACUS). Nine cases were negative on both urine and vaginal cytology (Table 2), including three with microscopic disease only at hysterectomy, three with atypical presentations and one where malignant cells were seen on re-review of the vaginal, but not urine, sample. There was no difference in detection rate according to grade, stage and histological subtype of endometrial cancer. Two patients with suspected endometrial cancer and positive vaginal cytology, including one whose urine cytology was also positive, were instead diagnosed with FIGO stage 1a and FIGO stage 3b high-grade serous adenocarcinomas of the fallopian tube, respectively. Another patient with suspected endometrial cancer and positive urine and vaginal cytology was diagnosed with a FIGO stage 1b poorly differentiated adenocarcinoma of the cervix on the final hysterectomy specimen. Positive cytology was more frequently observed in vaginal samples (endometrial cancer detection rate 90%) than urine samples (endometrial cancer detection rate 72%).

Of the 113 women with unexplained PMB, matched urine and vaginal samples were available for 91 women, vaginal samples

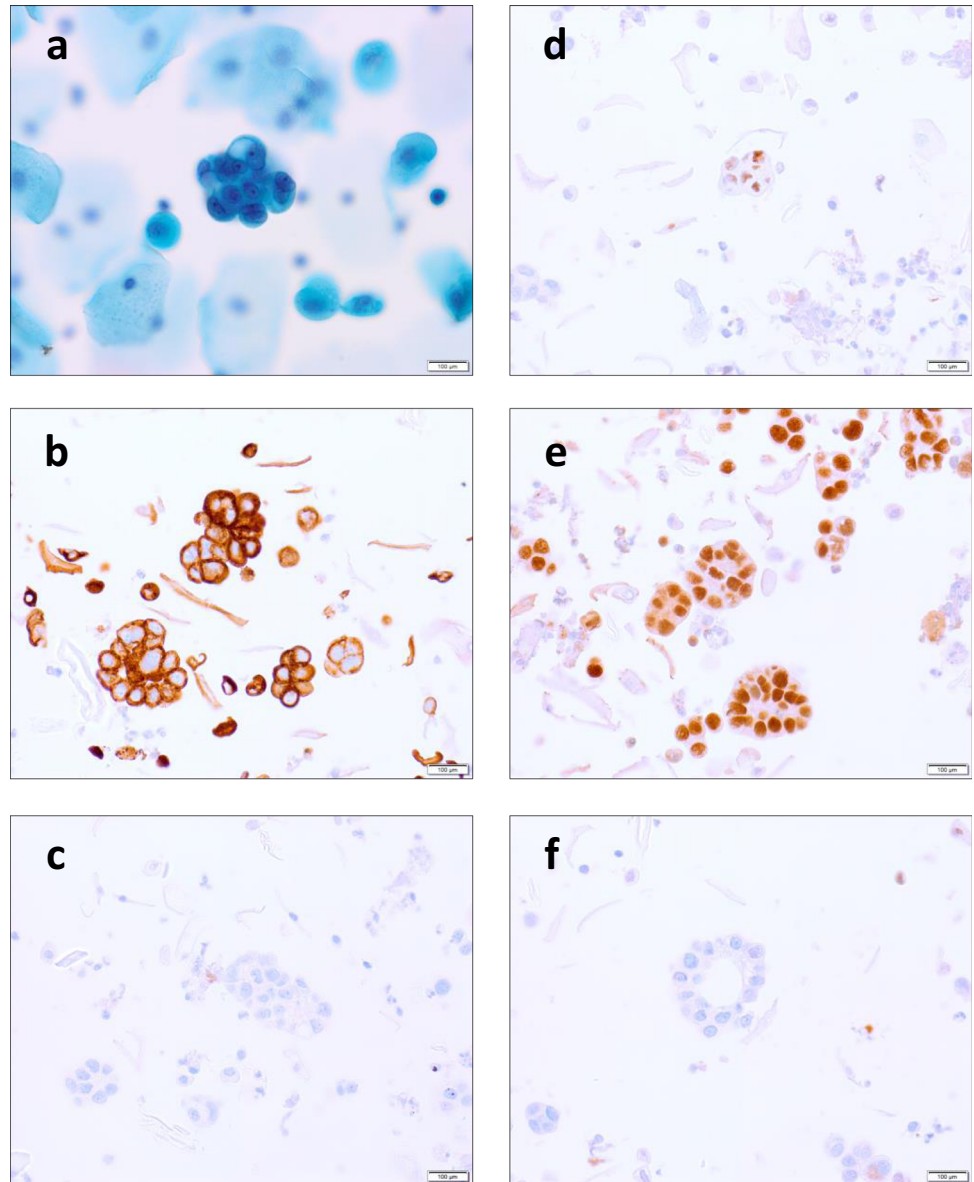

**Fig. 2 Photomicrograph showing morphology and immunohistochemistry profile of endometrial cancer cells in urogenital samples. a** Urine sample showing positive urine cytology (SurePath), demonstrating a single cluster of endometrial cells with malignant features, including enlarged hyperchromatic nuclei, coarse chromatin, prominent nucleoli, irregular nuclear outlines and high nuclear-to-cytoplasmic ratio. **b–f** Vaginal sample demonstrating the immunoprofile of these cells, consistent with epithelial cells from the female genital tract, CK7+ (**b**), CK20− (**c**), ER+ (**d**), PAX8+ (**e**), GATA3− (**f**). Scale bar = 100 μm.

only for 11 women and urine samples only for 11 women. All women underwent TVS; 70/113 had an endometrial thickness ≥4 mm. Further investigations with outpatient hysteroscopy (72.6%) and/or endometrial biopsy (74.34%) were completed secondary to a thickened endometrium or for clinical concern. Inadequate or failed investigations required repeat procedures in 36 women, including 8/113 (7.1%) where TVS failed to determine endometrial thickness, 11/113 (10%) where outpatient hysteroscopy failed or was abandoned, and 19 (17%) with failed or inadequate endometrial biopsy results. Repeat hysteroscopy under general anaesthesia was required for 17 women.

PMB was most commonly attributed to vulvovaginal atrophy (34.5%) or benign polyps (32.7%), but in a significant minority of cases, no underlying cause was found (18.6%) (Table 1). Six women with unexplained PMB were diagnosed with a gynaecological cancer, including four endometrial cancer (3 FIGO stage

1a grade 1/2 endometrioid endometrial cancers and a FIGO stage 3 carcinosarcoma), one cervical cancer (FIGO stage 4 squamous cell carcinoma) and an ovarian cancer (FIGO stage 3c high grade serous ovarian cancer). The patient with a stage 3 endometrial carcinosarcoma also had a synchronous bladder G3 TCC and CIS. A further three patients had atypical endometrial hyperplasia.

Urine and/or vaginal cytology detected all seven cancers in the PMB cohort. The ovarian cancer was only detected in urine. The patient with carcinosarcoma of the uterus and synchronous bladder cancer had two discrete subpopulations of malignant cells with distinct morphology in her samples. Subsequent immunostaining on formalin-fixed paraffin-embedded sections from agar cell blocks prepared from residual urine and vaginal samples found different immunoprofiles of these two malignant cell populations, consistent with one originating from the female

**Table 2 Women with endometrial cancer in whom cytology was false negative.**

| Patient ID | Age (years) | BMI (kg/m²) | Diagnosis | Details |
|---|---|---|---|---|
| DET03 | 78 | 36 | Stage 1a mixed clear cell and endometrioid EC | Degenerate urothelial cells observed on cytology |
| DET27 | 64 | 33 | Stage 3b G2 endometrioid EC | Atypical presentation (pelvic mass), no PMB, failed outpatient hysteroscopy and biopsy |
| DET51 | 69 | 34 | Stage 1a serous EC | Microscopic disease in hysterectomy specimen (focus measuring 1.7 mm) |
| DET55 | 54 | 32 | Stage 1a G1 endometrioid EC | Microscopic disease in hysterectomy specimen (foci measuring <2 mm combined) |
| DET160 | 56 | 41 | Stage 1a carcinosarcoma | Low volume, pauci-cellular vaginal sample |
| DET188 | 60 | 29 | Stage 1a G1 endometrioid EC | Microscopic disease in hysterectomy specimen (foci measuring 5 mm combined) |
| DET239 | 85 | 29 | Stage 1a clear cell EC | Atypical presentation (pelvic mass), no PMB, stenosed cervix |
| DET248 | 78 | 28 | Stage 1b G1 endometrioid EC | On cytology review, EC cells seen in vaginal cytology specimen, missed on first analysis |
| DET250 | 31 | 28 | Stage 1a G1 endometrioid EC | Atypical presentation, diagnosed through investigations for subfertility, no PMB. Macrophages only observed on review of vaginal cytology specimen. |

*BMI* body mass index, *EC* endometrial cancer, *PMB* postmenopausal bleeding.

**Table 3 Diagnostic accuracy of urogenital cytology for gynaecological cancer detection.**

| Test performance | Vaginal cytology % (95% CI) | Urine cytology % (95% CI) | Combined cytology[a] % (95% CI) |
|---|---|---|---|
| Sensitivity | 89.6 (82.2, 94.7) | 72.0 (62.1, 80.5) | 91.7 (84.9, 96.2) |
| Specificity | 88.7 (80.6, 94.2) | 94.9 (88.4, 98.3) | 88.8 (81.2, 94.1) |
| Positive likelihood ratio | 7.90 (4.5, 13.9) | 13.9 (5.3, 33.1) | 8.2 (4.8, 14.0) |
| Negative likelihood ratio | 0.12 (0.07, 0.21) | 0.30 (0.21, 0.41) | 0.09 (0.05, 0.17) |
| Disease prevalence | 52.2 (45.1, 59.3) | 50.8 (43.6, 57.9) | 50.5 (43.6, 57.3) |
| Positive predictive value | 89.7 (83.1, 93.8) | 93.5 (85.9, 97.5) | 89.3 (83.0, 93.4) |
| Negative predictive value | 88.7 (81.7, 93.2) | 76.7 (70.5, 81.9) | 91.4 (84.9, 95.2) |
| Diagnostic accuracy | 89.2 (84.1, 93.1) | 83.2 (77.3, 88.2) | 90.3 (85.5, 93.9) |

95% CI, 95% confidence intervals.
[a]Either vaginal or urine cytology positive.

genital tract (CK7+, PAX8+, ER+, CK20−, GATA3−), and the other from the urothelium (CK7+, PAX8−, ER−, CK20+, GATA3+). Three women were diagnosed with atypical hyperplasia. Only one was picked up by urine and/or vaginal cytology. One of the two missed cases was also missed on endometrial biopsy but subsequently diagnosed following therapeutic hysteroscopic endometrial polypectomy under general anaesthesia. Urine and/or vaginal cytology was positive for a further 12/107 cases where cancer was not diagnosed, giving a false-positive rate of 11.2%. Five women had ACUS detected in their urine and/or vaginal cytology, none of whom had endometrial cancer. Vaginal cytology was more accurate than urine cytology, picking up all four endometrial cancers (100%) for a 11.3% false-positive rate. Urine cytology identified cervical cancer in a woman who did not have a vaginal sample taken. Urine cytology detected two of the three endometrial cancers that provided a urine sample and also picked up the ovarian, bladder and cervical cancer, for a 5/98 (5.1%) false-positive rate.

Overall, combined vaginal and urine cytology had a sensitivity of 91.7% (95%CI 84.9%, 96.2%) and specificity of 88.8% (81.2%, 94.1%) for gynaecological cancer detection (Table 3). Vaginal cytology was more sensitive (90.2% versus 72.0%), but less specific (88.7% versus 94.9%) than urine cytology for endometrial cancer detection (difference in overall diagnostic accuracy, $p < 0.001$). For women in the prospective PMB cohort, combined vaginal and urine cytology had a sensitivity of 100% (95%CI 54.07%,100%) and specificity of 88.79% (95%CI 81.23%,94.07%) for gynaecological cancer detection.

**Inter-observer variability.** Overall, there were 12 cases where cytological analysis varied between the three observers (4 urine samples, 4 vaginal samples and 3 where both urine and vaginal cytology results differed). For urine samples, the Fleiss kappa coefficient was 0.962, $p < 0.01$ (95%CI 0.904–1.020) and for vaginal samples the Fleiss kappa coefficient was 0.951, $p < 0.01$ (95%CI 0.881–1.021).

**Acceptability of urine and vaginal sampling.** Over 94% of participants were familiar with collecting urine samples for medical tests. Mean pain scores for vaginal sampling (1.61, SD 2.04) were similar to those for TVS (1.68, SD 2.24, $p = 0.97$) but lower than for speculum examination (2.82, SD 2.88, $p < 0.001$), diagnostic hysteroscopy (4.28, SD 2.61, $p < 0.001$) and endometrial biopsy (4.88, SD 3.49, $p < 0.001$), respectively. There was no association between age, parity, history of pelvic pain or use of vaginal products and mean pain scores.

**Discussion**
This is the first report of a novel endometrial cancer detection tool that combines non-invasive urogenital sampling with cytology to differentiate malignant from non-malignant causes of PMB.

Exploiting the anatomical continuity between the uterine cavity and the lower genital tract, we demonstrate that malignant cells can be collected from the vagina by gentle lavage using the Delphi screener, and from voided urine samples that become contaminated with shed tumour debris during self-collection. These cells can be distinguished from benign squamous and urothelial cells by cytology where assessors are blinded to cancer outcomes. In our study of 216 women with PMB, urine and/or vaginal cytology was positive in 100/109 women who did (sensitivity 91.7%), and negative in 95/107 women who did not (specificity 88.8%) have cancer. In the prospective PMB cohort, urine and/or vaginal cytology detected all four endometrial cancers, offering comparable diagnostic accuracy to outpatient hysteroscopy and endometrial sampling[19]. These data suggest that urogenital cytology could offer a simple, patient-friendly 'rule out' test to enable quick reassurance for most women who present with PMB, and urgent invasive investigations for those who test positive.

Previous studies have shown that cytology brush samples taken from the uterine cavity are an effective, albeit invasive endometrial cancer detection tool[20–24]. A recent systematic review reported abnormal cervical cytology in 45% women with endometrial cancer, providing proof-of-concept that shed endometrial tumour cells can be collected from the cervix[14], and inspiring the development of novel sampling methodologies to improve cancer detection rates. Sample collection with the Delphi screener is simple, quick and patient-friendly[25], achieving lower mean pain scores than diagnostic hysteroscopy, endometrial biopsy and even speculum examination, which can be uncomfortable in elderly postmenopausal women[26]. It has the further advantage that it could be used by practice nurses and clinicians in community healthcare settings and even enable women to collect their own sample at a time and location convenient for them, extending its potential application to asymptomatic, high-risk women for whom no evidence-based screening tool currently exists. Using the Delphi screener, we missed 10/104 endometrial cancers in our study, including three with microscopic disease at hysterectomy and three with atypical presentations. The identification of patient, tumour and test parameters that increase the risk of false-negative cytology results must now be prioritized to establish the diagnostic limitations of this endometrial cancer detection tool.

There have been no studies investigating urine cytology for the detection of endometrial cancer, just three case reports of malignant cells in voided urine samples from women with localized cancer[27–29]. We found microscopic haematuria more frequently in cases than controls (77% versus 56%, $p = 0.04$), but the relationship with malignant cytology was not clear-cut. Urine had a sensitivity of 72.0%, which is inferior to vaginal cytology detection rates, but had the advantage of correctly identifying all but one of the tumours from other urogenital sites (specificity 94.9%), including a stage 3 ovarian cancer missed on vaginal cytology. Urine is a particularly attractive biofluid for endometrial cancer detection because it is straightforward to collect and highly acceptable to patients, offering opportunities for repeat sampling and collection at home[30].

Cytology is an established technology for the detection of cancers and their precursors, for example of the bladder[31], lung[32] and cervix[33]. Recent years have witnessed innovative solutions to the detection of Barrett's oesophagus[34] and oral cancer[35] using cytology, offering hope for early detection strategies[36]. Here we hijack the natural shed of endometrial tumours via PMB to similarly develop a novel approach to endometrial cancer detection. In the first report of its kind, we demonstrate proof-of-principle that vaginal fluid from women with endometrial cancer contains malignant cells that can be collected using minimally invasive sampling techniques and detected by cytology. Further, we show that specialist cytopathologists blinded to cancer outcomes can accurately distinguish women with and without sinister pathology by morphological assessment of urogenital cytospins. Indeed, in our cohort of women with unexplained PMB, cytology reliably identified cancers of the endometrium, cervix, bladder and ovary, when neither sample takers nor cytopathologists knew the results of standard diagnostics.

Limitations of our study include its single centre nature and relatively small numbers. We do not know whether the high diagnostic accuracy we observed is reproducible in other healthcare settings, particularly those without highly trained specialist cytopathologists. In contrast to cytopathologists, sample takers were only blinded to cancer outcomes in women being investigated for PMB, which may have influenced our results, although there was no difference in the quality of sample received by the cytopathologists, in terms of its volume or cellularity, according to final cancer status. A limitation of our approach is that it depends on the reliable shedding of endometrial tumours into the lower genital tract. We do not know whether urogenital cytology will be accurate for women with atypical presentations or in asymptomatic women. We identified only one of three cases of atypical hyperplasia. Premenopausal women made up just 8.3% of our study population in whom it may be challenging to distinguish malignant cells from benign mimics like polyps and cellular changes secondary to intrauterine devices. Adjunctive immunocytochemistry may help to define the malignant character and/or endometrial origin of cells[37,38] and work in this space is ongoing.

In conclusion, urogenital cytology could offer a simple, acceptable, easy to administer test that could be used in community settings as a triage tool for women with suspected endometrial cancer. Cytology positive women could be referred for diagnostic work-up while cytology negative women are quickly reassured without the need for unpleasant, invasive, anxiety-provoking tests, with substantial cost-saving implications for healthcare providers. Whilst our data are very promising, the clinical utility of urogenital cytology for endometrial cancer detection must now be confirmed in a large multicentre cross-sectional diagnostic accuracy study of women with unexplained PMB undergoing routine diagnostic investigations, with histology or clinical follow-up as the reference standard. Alternative diagnoses should be considered for participants with positive urogenital cytology but negative routine diagnostics.

## Methods

**Research ethics, approvals and patient involvement**. This study was approved by the North-West Greater Manchester Research Ethics Committee (reference-16/NW/0660) and all women gave written informed consent to participate. The study was sponsored by Manchester University NHS Foundation Trust and conducted in accordance with Good Clinical Practice guidelines and the Declaration of Helsinki. The research question was developed in partnership with patients, carers and healthcare professionals in the James Lind Alliance (JLA) Womb Cancer Priority Setting Partnership (PSP) (Question #2: Which women with abnormal vaginal bleeding should be referred urgently for investigations and which can be safely reassured?)[39], and endorsed by the JLA Detecting Cancer Early PSP (Question #1: What simple, non-invasive, painless, cost-effective, and convenient tests can be used to detect cancer early?)[40].

**Study participants**. We recruited women attending the Gynaecology Outpatient Department at St Mary's Hospital, Manchester University NHS Foundation Trust (MFT) between June-2016 and October-2018. Women referred for urgent investigation of unexplained PMB and those with known or suspected endometrial cancer awaiting hysterectomy were eligible to take part. PMB was defined as vaginal bleeding 12+ months after menstruation had stopped due to menopause. We excluded women undergoing investigations for abnormal pre-menopausal bleeding, in whom the risk of cancer is much lower, and those who had previously had a hysterectomy. In those women with known/suspected endometrial cancer, we included women of any age who presented with abnormal bleeding or other symptoms (e.g. haematuria, pelvic mass, abdominal pain), including those in whom routine clinical diagnostics failed, to estimate the accuracy of our approach in a 'real world' endometrial cancer population. We considered women who did not present with abnormal bleeding to have atypical presentations.

**Optimization of urine and vaginal sample collection**. First, we developed standardized protocols for urine collection and processing in a small pilot study. Urine samples collected and stored at 4 °C were stable for up to 48 h prior to processing. Best results were obtained when samples were fixed immediately after collection with an equal volume of CytoRich Red (CRR) (Becton, Dickinson and Company, New Jersey), to preserve cellular integrity, avoid bacterial overgrowth and lyse red blood cells, all of which hamper cytological interpretation. We compared cytological analysis of morning void urine samples with those collected in clinic at any time of the day, as well as the concordance of paired urine samples collected at different times of day. We found no difference in endometrial cancer detection rates according to the time of day samples were collected, but a quarter of 22 endometrial cancer patients had two urine samples had positive cytology on one and not the other, suggesting that two samples may boost detection rate. We trialled three different vaginal sampling methods: an introital brush (Evalyn® brush, Rovers Medical Devices), a cervical brush (Cervex-Brush®, Rovers Medical Devices) and the Delphi screener (Rovers Medical Devices), a CE-marked self-sampling tool. The introital and cervical brushes were inferior to the Delphi screener for endometrial cancer detection, possibly because the brushes collect cells from surfaces they are applied to directly while the Delphi screener uses gentle lavage to sample cervico-vaginal cells from a larger surface area. Collecting cells in a liquid phase may also yield higher quality samples than collecting them dry with a swab or a brush. Consistent with our findings, a systematic review of minimally invasive self-sampling methods for high-risk HPV detection found the Delphi screener collects four times as many cells as introital brushes[41]. The Delphi screener was therefore chosen for vaginal sampling in this study because of its superior performance for endometrial cancer detection, and its potential future use by healthcare practitioners in community settings and self-sampling by women themselves.

**Research sample collection**. Research samples were taken before any clinical procedures were carried out to avoid inadvertent contamination with iatrogenically dislodged endometrial cells. Urine was collected first to avoid uterine debris being removed from the lower genital tract by vaginal sampling. Voided urine samples were self-collected in a dry 60 ml urine collection pot and tested for haematuria by dipstick. Next, the patient was placed in the supine position with their legs bent and knees apart. The Delphi screener was inserted into the posterior fornix of the vagina and saline expelled from the reservoir. The liquid was re-aspirated by releasing the plunger whilst slowly rotating and retracting the device. A dry pot at the introitus collected any residual fluid. A second sample was obtained by re-filling the reservoir with saline and repeating the steps above.

**Acceptability of urine and vaginal sampling**. The acceptability of urine and vaginal sampling methodologies was assessed by asking women to rate the level of discomfort/pain experienced during speculum examination, TVS, endometrial biopsy and outpatient hysteroscopy using a 10-point visual analogue scale. We asked women whether they had previously provided a urine sample for a medical test. We asked if they were sexually active, whether they had experience of vaginal/pelvic pain conditions, and to reflect on their experience of vaginal products (e.g. use of tampons, contraceptive devices, dilators). The purpose of these questions was to determine the suitability of the Delphi screener to women with a range of different experiences, compared to standard clinical diagnostics for endometrial cancer detection.

**Clinical procedures**. For known or suspected endometrial cancer patients, samples were taken prior to hysterectomy in clinic or in the operating theatre at least 2 weeks after routine clinical diagnostics for endometrial cancer had been completed. For women with unexplained PMB, samples were taken in clinic immediately prior to routine diagnostic work-up, which included TVS followed by endometrial biopsy and outpatient hysteroscopy according to local guidelines. Briefly, women whose TVS showed a regular endometrial stripe <4 mm were considered at low risk of endometrial cancer and alternative diagnoses explored. Women with ≥4 mm endometrial thickness underwent further tests. A random biopsy using a pipelle endometrial sampler was indicated if TVS showed a regular, thickened endometrium. In the case of irregular thickening, where focal endometrial pathology was visualized or suspected, an outpatient hysteroscopy was performed and focal lesions biopsied under direct vision, followed by a pipelle endometrial biopsy. The acceptability of vaginal sampling, speculum examination, outpatient hysteroscopy and endometrial biopsy was compared using a pain visual analogue scale. Hysteroscopy was performed under general anaesthesia where outpatient hysteroscopy failed, biopsies were inadequate for diagnostic purposes or the procedure was poorly tolerated. Endometrial polyps were resected to allow full histological interpretation. Tissue samples were formalin-fixed, paraffin embedded, cut into 4-µm sections, stained with haematoxylin and eosin (H&E) and cover-slipped as per routine practice. At least one specialist gynaecological histopathologist reviewed all biopsies; suspicious or abnormal biopsies were reviewed by two specialist histopathologists according to FIGO-2009 staging criteria (endometrial cancer) and the WHO classification system (atypical hyperplasia)[42,43]; difficult cases were reviewed by additional specialist members of the gynaecological histopathology team.

**Cytological assessment of urine and vaginal samples**. Urine and vaginal samples were preserved with an equal volume of CRR (Becton, Dickinson and Company, New Jersey) and transferred to the Manchester Cytology Centre at MFT.

Once there, samples were centrifuged at 1614×g for 5 min. The supernatant was decanted and the pellet fixed in 6 ml CRR for 1 h. After fixation, the sample was centrifuged at 403×g for 10 min, supernatant decanted and the pellet prepared as a liquid-based cytology (LBC) Papanicalaou stained slide using the BD prepstain (Becton, Dickinson and Company, New Jersey) according to the manufacturer's instructions. The stained slide was dehydrated in two changes of industrial methylated spirits, cleared in two changes of xylene and cover-slipped. All slides were reviewed by three independent observers, a consultant biomedical scientist (BMS) in cytopathology and two consultant specialist cytopathologists, with discrepant cases settled by consensus review at a multi-header microscope. Slides were reported as unsatisfactory (sample obscured by debris, lymphocytes, bacteria or paucicellular sample), negative (benign urothelial and/or squamous epithelial cells) or positive (glandular, atypical or malignant cells seen). Positive cytology included atypical cells of undetermined significance (ACUS, occasional small atypical, degenerate cells), glandular cells (unexpected finding in postmenopausal woman) or malignant (single and/or clusters of epithelial cells with malignant features, specifically high nuclear-to-cytoplasmic ratio, vacuolated or minimal non-vacuolated cytoplasm, enlarged hyperchromatic nuclei, coarse chromatin, prominent nucleoli and irregular nuclear outlines).

**Blinding**. Women with unexplained PMB were sampled before clinical examination and any routine diagnostic tests, when researchers were blinded to cancer outcomes. The cytopathologists/BMS were blinded to patient identity, the results of clinical investigations and cancer outcomes until they had given their final report.

**Primary and secondary outcomes**. The pre-specified primary outcome was abnormal, suspicious or malignant glandular cells in urine and/or vaginal samples by cytology. Diagnostic accuracy was determined against routine histopathology on either the endometrial biopsy or hysterectomy specimen. Women with non-malignant causes of PMB were followed up for any subsequent referral or presentation to our service until May 2019; this was used as reference standard for women with no histopathology results.

**Sample size calculation**. A power calculation found that a sample size of 100 women with endometrial cancer and 100 women with PMB would enable the diagnostic accuracy of cytology to be estimated to ±7%, assuming sensitivity and specificity of ~85%. It was based on an estimated 5% endometrial cancer prevalence in women presenting to our service with PMB. An assumed sensitivity and specificity of 85% was considered the minimum acceptable diagnostic accuracy at which the test might be taken forward for validation. The ±7% relates to the 95% confidence interval surrounding our estimated sensitivity/ specificity.

**Statistical analysis**. Results were analysed using SPSS (version 22 IBM). Comparison of groups was analysed using a Chi-squared test. Overall diagnostic accuracy for cytology (sensitivity, specificity, positive predictive value and negative predictive value) was calculated using final clinical diagnosis as the reference standard. McNemar's test was used to compare the diagnostic accuracy of urine and vaginal cytology.

**Reporting summary**. Further information on research design is available in the Nature Research Reporting Summary linked to this article.

## Data availability

All data are available within the article, Source Data or available from the corresponding author upon request. Source data are provided with this paper.

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

## Acknowledgements

We would like to thank the women who participated in this study. We would like to thank the gynaecology, sonographer, cytopathology and histopathology staff who facilitated its completion. We are grateful to Dr. Catherine Fullwood who provided statistical support. This work was supported by the Medical Research Council [MR/M018431/1 to N.A.J.R.]; and the National Institute for Health Research (NIHR) [NIHR-CS-012-009 to E.J.C., IS-BRC-1215-20007 to E.J.C., DRF-2018-11-ST2-054 to H.O.F.]. The views expressed are those of the author(s) and not necessarily those of the NIHR or the Department of Health and Social Care.

## Author contributions

E.J.C. was Principal Investigator for the study and is its guarantor. E.J.C. designed the study and supervised its execution. H.O.F., N.A.J.R. and E.J.C. recruited women and collected the samples. N.N., D.S. and D.R. performed the cytological analyses. All authors interpreted the data. H.O.F. and E.J.C. wrote the manuscript. All authors provided critical comment, edited the manuscript, and approved its final version.

## Competing interests

The authors declare no competing interests.
