## [Peer Review File · Nature Communications]

Reviewers' Comments:

Reviewer #1:

Remarks to the Author:

This is a well-written paper reporting on a cross-sectional diagnostic accuracy study of urine and vaginal cytology specimens for detection of endometrial cancer. Using a cohort of 103 women with confirmed/suspected EC and 113 with unexplained postmenopausal bleeding (PMB), the sensitivity and specificity of urine and vaginal cytology combined were found to be 91.7% and 88.8%, respectively.

Although the subject and the findings are of relevance, some aspects of the study and the analysis should be clarified.

Introduction:

1. The reasoning, aim and methodology of the study do not seem to completely align. First the authors state that postmenopausal bleeding would be the presenting symptom for women to be screened by the new method: See introduction: "We hypothesized that PMB offers a potential liquid biopsy to enable EC detection in lower urogenital tract samples. The aim of this study was to establish the diagnostic potential of voided urine and non-invasive vaginal sampling for the detection of EC by cytology." (page 4, lines 68-70). This hypothesis is mentioned again in their discussion section: "Here we hijack the natural shed of endometrial tumours via postmenopausal bleeding to similarly develop a novel approach to EC detection" (page 10, lines 235-236). An additional benefit of this approach would be the avoidance of samples from premenopausal women, as these tend to be more challenging for the cytopathologist to diagnose. However, later on the authors describe they have included patients that did not present with PMB (over 20% of the cohort) and have even included premenopausal women (8.3%). I assume this has been done in order to enrich the cohort with EC patients. However, it results in an study population that will not fully address the aim of the manuscript. Please, clarify aim and methods.

Methods:

1. To have a better estimate of the possible difficulty of the diagnosis of EC using cytology, data regarding the number of cases that had a discrepant diagnosis among the three observers would be useful. Furthermore, the observers seemed to be highly specialized (consultant specialist cytopathologists and a consultant biomedical scientist in cytopathology) working in a tertiary referral centre – could the authors comment on the level of specific training required? Could community center cytology perform the evaluation with the same diagnostic accuracy?

2. Could the authors provide the number of women with a cytology result of atypical cells of undetermined significance (ACUS) and glandular cells that were ultimately diagnosed as EC and those that didn't. Again, it would inform on the reliability of cytological diagnosis.

3. The authors describe specifically the Delphi screener for taking the vaginal samples – is this the only possible technique? Could it be done by a GP or even by the patient herself? Some specification in methods and/or discussion would be helpful.

4. Statistical methods:

I. The sample size calculation is only stated as 'a sample size of 200 would enable the diagnostic accuracy to be +/-7% (is the SD of the accuracy meant here?), assuming sensitivity and specificity of 85%'. On what data these assumptions have been based is unclear.

II. Acceptability seems to be investigated only by assessing pain on a visual analogue scale (for part of the participants) – the definition and analysis of acceptability are not mentioned in the methods

III. Considering women without PMB as well as premenopausal women have been included in the study, has this affected the sample size calculation?

Results:

1. In table 1, the authors show in the second column depicting patients with PMB that 3.5% of the patients had "other" presenting complaints than PMB, how can this be? Were these patients not by definition presenting with PMB?

Discussion:

1. The difficulty of EC diagnosis, as the authors point out, lies in the differential diagnosis with benign mimics in premenopausal women. Only 8.3% of the patients were premenopausal (18 patients) and 3 of them had a false negative cytology, resulting in a lower sensitivity as compared to the general cohort. A more detailed information on the results regarding premenopausal women would help to understand the possible limitations of endometrial cytology. Also, the diagnosis of AH was not easy with cytology (1 in 3), while complex AH is usually an indication for hysterectomy in postmenopausal patients.

2. Page 9, lines 215-216, the authors state: "Using the Delphi screener, we missed just 9/109 EC cases in our study," Considering this is meant to be a screening method for EC detection, the misdiagnosing of 8% of the patients is quite high, especially considering the problematic population for diagnosis (premenopausal women) has not been included in the study. Please, rephrase. Would the authors recommend this tool to be used before or instead of TVS, or as only diagnostic method in women with PMB?

3. Page 10, lines 236-238: "In the first report of its kind, we demonstrate proof of principle that vaginal fluid contains malignant cells in women with EC." The existence of endometrial cells in vaginal and cervical cytology specimens has been known for a long time (and studies have been published, refs 11-14 for example). The authors could stress the novelty of their clinical study and the high yield of liquid cytology using their techniques more specifically.

4. General comment regarding the discussion: this section is rather long, and there are sections which would belong to or repeat the introduction or methods: for example, the first and especially the second paragraph both describe the 'gentle lavage technique using the Delphi screener and how it was designed' (belongs to methods, here only the patient-friendliness (data?) and use (also for GP? self-sampling?) should be discussed); the 3rd paragraph describes storage conditions etc. The discussion could be condensed and have a clear conclusion ('clinical utility must now be confirmed', please specify: should this be a randomised trial? Comparing cytology against with current standard?)

Reviewer #2:

Remarks to the Author:

I would like to thank the editorial office for giving me the opportunity to review the manuscript submitted by O'Flynn and colleagues examining the diagnostic accuracy of cytology for the detection of endometrial cancer in urine and vaginal samples.

This is an important topic and I commend the efforts for exploring this therapeutic opportunity in clinical care. The authors report that combination urine and vaginal cytology, using the Delphi screener, had a sensitivity of 91.7%, specificity of 88.8%, with 11.2% false negative rate.

Although this is a proof of principle study there are several important questions and limitations detailed below.

The authors claim that the sensitivity and specificity noted above are similar to that seen with office pipelle based EMB or operative theater based D&C. However, existing literature suggests that the sensitivity of EMB in women with postmenopausal bleeding approaches 100%, with specificity of 98% (Alam et al. and Heintz et al.).

On page 4, line 74 the authors state 100 women with known or suspected EC, while on page 4, line 89, they refer to 103 women with known or suspected EC. Why the discrepancy between 100 and 103?

Can the authors please define "known or suspected EC". How was this determined aside from the presence of PMB? Did these patients have alternate procedures, imaging, gross pathology on

clinical exam. If so, could this impact the accuracy of the assays being examined?

Page 5, line 103. The authors report that women with EMS < 4mm did not require endometrial assessments. As published by Wang et al, in 2006 EMS alone can miss serous and clear cell histology. How are women with EMS < 4mm, with persistent PMB managed?

Page 6, lines 125-126: what is meant by the phrase "atypical presentation"?

Page 6, line 132: How would this assay distinguish between an endocervical adenocarcinoma and an endometrial adenocarcinoma given the different management strategies? Is this based on IHC alone, HPV in situ hybridization?

It is unclear to me what the urine assessment adds to the detection of endometrial cancer. Were any cases detected on urine alone? I understand that urine cytology identified alternate cancers, and do not understand the biologic rationale in those cases, but I am not certain from the data presented that urine cytology adds anything to endometrial cancer detection?

Page 7, line 140: How was stage 4 cervical cancer not detected on clinical exam? This should not be described as "unexplained" PMB.

Page 7, line 165: What is meant by the phrase, "routine clinical diagnosis...?"

How do the authors suggest providers manage + cytology (with the methods described in the paper), with negative EMB in clinical practice? With an 11.2% false + rate, which is much higher than endometrial biopsy and D&C, would providers ignore the cytology? Not sure how this would be managed clinically.

The authors do not discuss the ability of the vaginal or urine cytology to accurately detect or inform underlying histology. The presence of serous or clear cell carcinoma vs endometrioid histology may have implications in patient counseling, surgical staging and need for preoperative imaging. This needs to be discussed and addressed as part of the manuscript.

Response to reviewers' comments

Reviewer comment	Author response
Reviewer 1 Comment 1	Reviewer 1 Response 1
This is a well-written paper reporting on a cross-sectional diagnostic accuracy study of urine and vaginal cytology specimens for detection of endometrial cancer. Using a cohort of 103 women with confirmed/suspected EC and 113 with unexplained postmenopausal bleeding (PMB), the sensitivity and specificity of urine and vaginal cytology combined were found to be 91.7% and 88.8%, respectively. Although the subject and the findings are of relevance, some aspects of the study and the analysis should be clarified.	Thank you.
Reviewer 1 Comment 2	Reviewer 1 Response 2
Introduction: The reasoning, aim and methodology of the study do not seem to completely align. First the authors state that postmenopausal bleeding would be the presenting symptom for women to be screened by the new method: See introduction: "We hypothesized that PMB offers a potential liquid biopsy to enable EC detection in lower urogenital tract samples. The aim of this study was to establish the diagnostic potential of voided urine and non-invasive vaginal sampling for the detection of EC by cytology." (page 4, lines 68-70). This hypothesis is mentioned again in their discussion section: "Here we hijack the natural shed of endometrial tumours via postmenopausal bleeding to similarly develop a novel approach to EC	We hypothesized that tumour shed provides an opportunity to detect EC by non-invasive sampling. To demonstrate proof of principle that urogenital cytology can detect EC, we enriched our cohort by including women with known/suspected EC, irrespective of their menopausal status. Premenopausal women with abnormal bleeding rarely get referred for specialist investigations via a suspected cancer pathway at our centre, but rather get diagnosed from a variety of non-specialist gynaecology clinics. This not only makes access to this subset of women challenging but also risks sampling many hundreds of women to achieve a single

detection” (page 10, lines 235-236). An additional benefit of this approach would be the avoidance of samples from premenopausal women, as these tend to be more challenging for the cytopathologist to diagnose. However, later on the authors describe they have included patients that did not present with PMB (over 20% of the cohort) and have even included premenopausal women (8.3%). I assume this has been done in order to enrich the cohort with EC patients. However, it results in a study population that will not fully address the aim of the manuscript. Please, clarify aim and methods.

case of EC, which was not feasible for this proof of concept study. If we had recruited any woman with abnormal vaginal bleeding, the average age of the cancer cases and control groups in this study would have been vastly different, prompting criticism that it was not an appropriately matched control group. We considered it more important to match for age and the most common demographic (90% cases are in postmenopausal women presenting with PMB) than to have very large numbers of premenopausal women in whom the risk of EC would be very low.

Premenopausal women are likely to create a diagnostic challenge for cytopathology assessment, given the number of benign mimics that might be observed. Including them in our EC cohort therefore improves the reliability of our findings.

We understand that these decisions may be confusing to the reader so have added an explanation for this in the methods section:

We excluded women undergoing investigations for abnormal premenopausal bleeding, in whom the risk of cancer is much lower, and those who had previously had a hysterectomy. In those women with known/ suspected EC, we included women of any age who presented with abnormal bleeding or other symptoms (eg haematuria, pelvic mass, abdominal pain), including those in whom routine clinical diagnostics failed, to estimate the accuracy of our approach in a ‘real world’ EC population. We considered women who did not present

	with abnormal bleeding to have atypical presentations.
Reviewer 1 Comment 3	Reviewer 1 Response 3
Methods: To have a better estimate of the possible difficulty of the diagnosis of EC using cytology, data regarding the number of cases that had a discrepant diagnosis among the three observers would be useful. Furthermore, the observers seemed to be highly specialized (consultant specialist cytopathologists and a consultant biomedical scientist in cytopathology) working in a tertiary referral centre – could the authors comment on the level of specific training required? Could community center cytology perform the evaluation with the same diagnostic accuracy?	Overall, there were 12 cases where cytological analysis varied (4 urine samples, 4 vaginal samples and three where both urine and vaginal cytology results differed between observers). For urine samples, the Fleiss kappa coefficient was 0.962 and for vaginal samples the Fleiss kappa coefficient was 0.951. We have added the following section to the results: Inter-observer variability Overall, there were 12 cases where cytological analysis varied between the three observers (4 urine samples, 4 vaginal samples and 3 where both urine and vaginal cytology results differed). For urine samples, the Fleiss kappa coefficient was 0.962, $p < 0.01$ (95%CI 0.904-1.020) and for vaginal samples the Fleiss kappa coefficient was 0.951, $p < 0.01$ (95%CI 0.881-1.021). In the UK, consultant cytopathologists undergo in-depth specialist training and develop expertise in cytological analysis through exposure to a large number of cases. To practice at a cancer referral centre (like ours), the training requirement is the FRCPath (part 2) or equivalent. For Consultant Biomedical Scientists, the requirement is the IBMS/RCPPath advanced specialist diploma in non-gynaecological cytology. Whilst we can see no reason why community centre cytopathologists would not be able to achieve the same diagnostic

	standards as their teaching hospital colleagues, we accept that they may not have access to the required training and sample throughput to equip them with the necessary diagnostic skills. We note that this is the case for any specialised cytology service rather than being a unique challenge for EC diagnosis (for example, in the UK, cervical cytology is now ‘centralised’ to 12 cervical screening laboratories nationwide). We have added a comment to this effect in the discussion: We do not know whether the high diagnostic accuracy we observed is reproducible in other healthcare settings, particularly those without highly trained specialist cytopathologists.
Reviewer 1 Comment 4	Reviewer 1 Response 4
Methods: Could the authors provide the number of women with a cytology result of atypical cells of undetermined significance (ACUS) and glandular cells that were ultimately diagnosed as EC and those that didn’t. Again, it would inform on the reliability of cytological diagnosis.	In the PMB cohort, 5 women had ACUS on urogenital cytology, none of whom had cancer. In the EC cohort, 2 women had ACUS on urogenital cytology. We have added the following: Urine and/or vaginal cytology was positive in 94 of 103 cases, of which two were reported as showing abnormal cells of unknown significance (ACUS). And Five women had ACUS detected in their urine and/or vaginal cytology, none of whom had EC.

Reviewer 1 Comment 5	Reviewer 1 Response 5
Methods: The authors describe specifically the Delphi screener for taking the vaginal samples – is this the only possible technique? Could it be done by a GP or even by the patient herself? Some specification in methods and/or discussion would be helpful.	The Delphi screener was designed as a self-sampling tool for HPV primary cervical screening; it has been tested and CE marked for this purpose. It is easy to use and simple to understand. The screener comes in a single use pack with written and pictorial instructions that can be understood by patients and clinicians from all settings, including GPs and practice nurses. Indeed, this is the reason why we chose this device for our study. Its intended use is the community setting, where we hope it could be used by GPs and practice nurses as a rule in/ rule out test for specialist referral for symptomatic women, and with the ultimate ambition of rolling it out as a screening tool for high risk women to use in the comfort of their own homes and return to the lab by post or via their local GP practice. Other lower genital tract sampling devices, eg the cervical broom (used in routine cervical screening) and any of a range of different vaginal swabs, may also be suitable for obtaining endometrial cancer cells for cytological assessment. We trialled cervical and introital brushes in a small cohort of our study population and found a sensitivity of only 30%. We hypothesized that the Delphi screener sampled a much larger surface area of the vagina and cervix than either of these other tools, thus explaining the discrepant results. Collecting cells in a liquid phase may also yield higher quality samples than collecting them dry with a swab or a brush. A systematic review of minimally-invasive self-sampling methods for high risk HPV detection found the Delphi screener obtained a four-fold higher cell

yield than brush-based cell collection, consistent with our findings (Snijders et al, 2013).

We have added the following to the methods:

We trialed three different vaginal sampling methods: an introital brush (Evalyn® brush, Rovers Medical Devices), a cervical brush (Cervex-Brush®, Rovers Medical Devices), and the Delphi screener (Rovers Medical Devices), a CE-marked self-sampling tool. The introital and cervical brushes were inferior to the Delphi screener for EC detection, possibly because the brushes collect cells from surfaces they are applied to directly while the Delphi screener uses gentle lavage to sample cervico-vaginal cells from a larger surface area. Collecting cells in a liquid phase may also yield higher quality samples than collecting them dry with a swab or a brush. Consistent with our findings, a systematic review of minimally-invasive self-sampling methods for high risk HPV detection found the Delphi screener collects four times as many cells as introital brushes⁴⁰. The Delphi screener was therefore chosen for vaginal sampling in this study because of its superior performance for EC detection, and its potential future use by healthcare practitioners in community settings and self-sampling by women themselves.

And in the discussion, it says:

Sample collection with the Delphi screener is simple, quick and patient-friendly²⁵, achieving lower mean pain scores than diagnostic hysteroscopy, endometrial biopsy and even speculum examination,

	which can be uncomfortable in elderly postmenopausal women²⁶. It has the further advantage that it could be used by practice nurses and clinicians in community healthcare settings and even enable women to collect their own sample at a time and location convenient for them, extending its potential application to asymptomatic, high-risk women for whom no evidence-based screening tool currently exists.
Reviewer 1 Comment 6	Reviewer 1 Response 6
Statistical methods: I. The sample size calculation is only stated as 'a sample size of 200 would enable the diagnostic accuracy to be +/-7% (is the SD of the accuracy meant here?), assuming sensitivity and specificity of 85%'. On what data these assumptions have been based is unclear.	I. The power calculation was based upon a presumed prevalence of endometrial cancer of 5% in a cohort of women presenting to our service with postmenopausal bleeding. An assumed sensitivity and specificity of 85% was considered the minimum acceptable diagnostic accuracy at which the test might be taken forward for validation. +/-7% relates to the 95% confidence interval, which would extend 7% either side of the estimate, assuming a sensitivity of around 85%. This means that our study was sufficiently powered to estimate sensitivity/specificity of the test to a margin of 7% either side of 85%. In other words, the confidence intervals give a measure of the precision with which we can estimate test performance in a study of this size. We have clarified these assumptions in the methods: Sample size calculation A power calculation found that a sample size of 100 women with EC and 100 women with PMB would enable the diagnostic accuracy of cytology to be estimated to +/-7%, assuming sensitivity and specificity

II. Acceptability seems to be investigated only by assessing pain on a visual analogue scale (for part of the participants) – the definition and analysis of acceptability are not mentioned in the methods

of ~85%. It was based on an estimated 5% EC prevalence in women presenting to our service with PMB. An assumed sensitivity and specificity of 85% was considered the minimum acceptable diagnostic accuracy at which the test might be taken forward for validation. The +/-7% relates to the 95% confidence interval surrounding our estimated sensitivity/ specificity.

II. As pain is the leading cause of failure of outpatient hysteroscopy and endometrial biopsies, we wanted to see how users of the new test thought it compared to routine tests in terms of how painful they are. We accept that pain is a crude measurement of test acceptability and, going forwards, we plan a more in-depth exploration of acceptability through a qualitative study based on the work of Sekhon et al (3). This study is currently in set up phase. We have added further detail in the methods section about this:

Acceptability of urine and vaginal sampling

The acceptability of urine and vaginal sampling methodologies was assessed by asking women to rate the level of discomfort/ pain experienced during speculum examination, TVS, endometrial biopsy and outpatient hysteroscopy using a 10-point visual analogue scale. We asked women whether they had previously provided a urine sample for a medical test. We asked if they were sexually active, whether they had experience of vaginal/pelvic pain conditions, and to reflect on their experience of vaginal products (eg use of tampons, contraceptive devices, dilators). The purpose of these questions was

III. Considering women without PMB as well as premenopausal women have been included in the study, has this affected the sample size calculation?	to determine the suitability of the Delphi screener to women with a range of different experiences, compared to standard clinical diagnostics for EC detection. III. The sample size was based on an estimated 5% prevalence of EC in a PMB cohort. The known EC cases were a consecutive sample of women diagnosed with EC in our department, irrespective of age, menopausal status, presenting symptoms and success of routine diagnostics. If we had restricted known cases of EC to those with postmenopausal bleeding in whom routine diagnostics were successful, we would have biased our study towards the 'easy' diagnoses and missed the opportunity to estimate test performance in a 'real world' cohort of cases. We didn't want to over-estimate test performance in this way, hence the 'mixed' cohort of EC cases. But, as above, we think this strengthens our assessment of the performance of the new test rather than detracts: despite including some tricky cases, the test still had 92% sensitivity and 89% specificity.
Reviewer 1 Comment 7	Reviewer 1 Response 7
Results: In table 1, the authors show in the second column depicting patients with PMB that 3.5% of the patients had "other" presenting complaints than PMB, how can this be? Were these patients not by definition presenting with PMB?	These four women presented with postcoital bleeding. On reflection, we agree this is confusing and have now included them within the PMB group. They presented with abnormal vaginal bleeding after the menopause; there is no need to distinguish postcoital from 'regular' postmenopausal bleeding as we had done. Thank you for pointing this out.

	We have similarly changed the ‘other’ presenting complaint in the known cancer group, moving women with postcoital bleeding after the menopause to the PMB group, clarifying the number of women with abnormal pre-/ peri-menopausal bleeding, and adding a table footnote to describe the atypical presentations in more detail.
Reviewer 1 Comment 8	Reviewer 1 Response 8
Discussion: The difficulty of EC diagnosis, as the authors point out, lies in the differential diagnosis with benign mimics in premenopausal women. Only 8.3% of the patients were premenopausal (18 patients) and 3 of them had a false negative cytology, resulting in a lower sensitivity as compared to the general cohort. A more detailed information on the results regarding premenopausal women would help to understand the possible limitations of endometrial cytology. Also, the diagnosis of AH was not easy with cytology (1 in 3), while complex AH is usually an indication for hysterectomy in postmenopausal patients.	The majority of EC diagnoses were in postmenopausal women (>90%), which is consistent with the literature. We didn’t select EC cases based on any clinical features, specifically age, menopausal status, symptoms or the success or otherwise of routine diagnostics, in order to reduce selection bias. We avoided premenopausal women in the prospective PMB cohort in order to observe a 5% frequency of EC within that group; if we had included premenopausal women with abnormal bleeding, we would have needed to sample many hundreds to achieve one EC diagnosis. Given that this was a proof of principle study, we did not think this was a good use of resources. In the premenopausal women with EC, only one had negative urogenital cytology (Table 2), and we have highlighted this further in the results section. We agree that AH would be an indication for hysterectomy, however, our study was designed to assess the diagnostic potential of urogenital cytology for EC diagnosis and we only had three cases of AH in the study. One was detected, one was missed both by

	urogenital cytology and endometrial biopsy (routine diagnostics) and one was missed by urogenital cytology but identified by pipelle endometrial biopsy. There are insufficient numbers of premenopausal women and cases of AH to determine the utility of urogenital cytology in these groups, and further research is needed. We have focused our study on the potential utility of urogenital cytology for the detection of EC in postmenopausal women, in whom 90% cases are diagnosed. We have highlighted this limitation in the discussion: Premenopausal women made up just 8.3% of our study population in whom it may be challenging to distinguish malignant cells from benign mimics like polyps and cellular changes secondary to intrauterine devices. Also: We identified only one of three cases of atypical hyperplasia. And: The identification of patient, tumour and test parameters that increase the risk of false negative cytology results must now be prioritised to establish the diagnostic limitations of this EC detection tool.
Reviewer 1 Comment 9	Reviewer 1 Response 9

Discussion:

Page 9, lines 215-216, the authors state: “Using the Delphi screener, we missed just 9/109 EC cases in our study,” Considering this is meant to be a screening method for EC detection, the misdiagnosing of 8% of the patients is quite high, especially considering the problematic population for diagnosis (premenopausal women) has not been included in the study. Please, rephrase. Would the authors recommend this tool to be used before or instead of TVS, or as only diagnostic method in women with PMB?

Thank you for raising this important point.

We do not see urogenital cytology replacing standard diagnostics for high risk women and/or those with recurrent symptoms. We see it being used in community settings to identify symptomatic women for fast track referral or reassurance. The vast majority of women with PMB do not have EC yet all women currently undergo specialist referral, a series of invasive, painful and expensive tests and the psychological and physical consequences of these.

The Delphi screener missed 9/109 EC cases but there were robust explanations for 6/9 of the missed cases (Table 2). Three presented with a pelvic mass and no PMB. Urogenital cytology relies on the anatomical continuity between the uterine cavity and the lower genital tract, and is less likely to be accurate in women with cervical stenosis in whom bleeding is not the presenting symptom. To be clear, we do not see the Delphi screener being used for the detection of EC in women who do NOT present with PMB.

Three of the missed 9 were diagnosed through the hysteroscopic resection of a polyp but when they were sampled for this study (immediately prior to hysterectomy), there was only microscopic disease (<5mm) left in the uterus. It is unlikely that a lesion <5mm in maximum dimension will be shedding sufficient numbers of malignant cells to be identified by urogenital cytology.

	One of 9 was 'missed' on first cytology review and malignant EC cells only identified on second review, when the diagnosis was revealed to the cytopathology team. This study provides proof of principle but future work must focus on identifying which women with EC that urogenital cytology is useful for and which women it is not. We have re-worded as requested: Using the Delphi screener, we missed 9/109 EC cases in our study, including three with microscopic disease at hysterectomy and three with atypical presentations. The identification of patient, tumour and test parameters that increase the risk of false negative cytology results must now be prioritised to establish the diagnostic limitations of this EC detection tool.
Reviewer 1 Comment 10	Reviewer 1 Response 10
Discussion: Page 10, lines 236-238: "In the first report of its kind, we demonstrate proof of principle that vaginal fluid contains malignant cells in women with EC." The existence of endometrial cells in vaginal and cervical cytology specimens has been known for a long time (and studies have been published, refs 11-14 for example). The authors could stress the novelty of their clinical study and the high yield of liquid cytology using their techniques more specifically.	Thank you, you are correct. We have amended this statement in the discussion to read: In the first report of its kind, we demonstrate proof of principle that vaginal fluid from women with EC contains malignant cells that can be collected using minimally-invasive sampling techniques and detected by cytology.

Reviewer 1 Comment 11	Reviewer 1 Response 11
General comment regarding the discussion: this section is rather long, and there are sections which would belong to or repeat the introduction or methods: for example, the first and especially the second paragraph both describe the ‘gentle lavage technique using the Delphi screener and how it was designed’ (belongs to methods, here only the patient-friendliness (data?) and use (also for GP? self-sampling?) should be discussed); the 3rd paragraph describes storage conditions etc. The discussion could be condensed and have a clear conclusion (‘clinical utility must now be confirmed’, please specify: should this be a randomised trial? Comparing cytology against with current standard?)	Thank you for this. We have restructured the discussion and moved methods and results to their appropriate sections. We have outlined that a cross sectional prospective multicentre trial should be performed next. We have re-written the conclusion to give a clearer message: In conclusion, urogenital cytology could offer a simple, acceptable, easy to administer test that could be used in community settings as a triage tool for women with suspected EC. Cytology positive women could be referred for diagnostic work up while cytology negative women are quickly reassured without the need for unpleasant, invasive, anxiety-provoking tests, with substantial cost-saving implications for healthcare providers. Whilst our data are very promising, the clinical utility of urogenital cytology for EC detection must now be confirmed in a large multicentre prospective study of women with unexplained PMB undergoing investigation.
Reviewer 2 Comment 1	Reviewer 2 Response 1
I would like to thank the editorial office for giving me the opportunity to review the manuscript submitted by O’Flynn and colleagues examining the diagnostic accuracy of cytology for the detection of endometrial cancer in urine and vaginal samples. This is an important topic and I commend the efforts for exploring this therapeutic opportunity in clinical care. The authors report that combination urine and vaginal cytology, using the Delphi screener,	Thank you.

had a sensitivity of 91.7%, specificity of 88.8%, with 11.2% false negative rate.	
Reviewer 2 Comment 2	Reviewer 2 Response 2
Although this is a proof of principle study there are several important questions and limitations detailed below. The authors claim that the sensitivity and specificity noted above are similar to that seen with office pipelle based EMB or operative theater based D&C. However, existing literature suggests that the sensitivity of EMB in women with postmenopausal bleeding approaches 100%, with specificity of 98% (Alam et al. and Heintz et al.).	Thank you for raising this important point. Whilst the Delphi screener missed 9 known EC cases, there were strong mitigating circumstances for 6 of these, including 3 with atypical presentations (no abnormal bleeding) and 3 with microscopic disease only at hysterectomy. We would like to emphasise that in our prospective PMB cohort, urogenital cytology had 100% sensitivity for the detection of EC. It also picked up three other cancers (of the cervix, ovary and bladder). Few diagnostic tests in established clinical practice have a sensitivity >90%. As the comparator technology, endometrial biopsy has high overall accuracy in diagnosing EC when an adequate specimen is obtained. Several high quality studies and systematic reviews have investigated the diagnostic accuracy of endometrial biopsy and find that the procedure failure rate and the non-diagnostic/inadequate pathology rate are as high as 7-11% and 15-76%, respectively (Clark et al, BJOG, 2002; Visser et al, Obstet Gynecol, 2017; van Hanegem et al, Eur J Obstet Gynecol Reprod Biol, 2016). When the procedure fails in the outpatient setting or the tissue sample is non-diagnostic (approx. 30% according to systematic reviews above), repeat procedures under general anaesthetic are recommended - a waste of resources and at considerable personal cost to women if they are at low risk of cancer

	in the first place. Alam et al compared the diagnostic potential of pipelle vs dilatation and curettage whilst women were under general anaesthetic in the operating theatre, not in the outpatient setting. They included only 11 women with PMB of whom only 2 had endometrial cancer. Heintz et al reported a high sensitivity and specificity for pipelle but there was an 11% failure rate. We have moderated the sentence the reviewer refers to: In our study of 216 women with PMB, urine and/or vaginal cytology was positive in 100/109 women who did (sensitivity 91.7%), and negative in 95/107 women who did not (specificity 88.8%) have cancer. In the prospective PMB cohort, urine and/or vaginal cytology detected all 4 EC, offering comparable diagnostic accuracy to outpatient hysteroscopy and endometrial sampling¹⁹.
Reviewer 2 Comment 3	Reviewer 2 Response 3
On page 4, line 74 the authors state 100 women with known or suspected EC, while on page 4, line 89, they refer to 103 women with known or suspected EC. Why the discrepancy between 100 and 103?	Thank you for spotting this typo, which we have now corrected. It should read 103 throughout.
Reviewer 2 Comment 4	Reviewer 2 Response 4
Can the authors please define “known or suspected EC”. How was this determined aside from the presence of PMB? Did these patients have	The women with known or suspected EC had a tissue diagnosis or highly suspicious findings on scan. We do not think this influenced the

alternate procedures, imaging, gross pathology on clinical exam. If so, could this impact the accuracy of the assays being examined?	performance of our test because they were sampled 2+ weeks after their routine diagnostics had been completed, whilst under general anaesthetic for their hysterectomy, immediately pre-op in the clinic room or immediately before starting treatment with progestins. To clarify this, we have added the following sentences to the methods: Research samples were taken before any clinical procedures were carried out to avoid inadvertent contamination with iatrogenically-dislodged endometrial cells. Urine was collected first to avoid uterine debris being removed from the lower genital tract by vaginal sampling. Also: For known or suspected EC patients, samples were taken prior to hysterectomy in clinic or in the operating theatre at least 2 weeks after routine clinical diagnostics for EC had been completed.
Reviewer 2 Comment 5	Reviewer 2 Response 5
Page 5, line 103. The authors report that women with EMS < 4mm did not require endometrial assessments. As published by Wang et al, in 2006 EMS alone can miss serous and clear cell histology. How are women with EMS < 4mm, with persistent PMB managed?	As per guidance in the UK from the British Gynaecological Cancer Society (BGCS), women with an endometrial thickness of under 4mm are not further evaluated on first presentation. If the patient returns with recurrent symptoms, they are offered hysteroscopy and endometrial biopsy. We are aware that diagnoses are occasionally missed at an ET<4mm,

	but this is uncommon. BGCS guidelines recommend an endometrial thickness cut-off of ≥ 4 mm based on a systematic review published in 2010, which included 13 studies of 2,896 patients with PMB of whom 259 were diagnosed with EC. A cut-off of ≥ 4 mm had a sensitivity of 94.8% (95%CI 86.1–98.2%) and a specificity of 46.7% (95%CI 38.3–55.2%) for EC detection (Timmermans et al, 2010).
Reviewer 2 Comment 6	Reviewer 2 Response 6
Page 6, lines 125-126: what is meant by the phrase “atypical presentation”?	‘Atypical presentation’ means those women who did not present with abnormal bleeding. A high quality systematic review concluded that 91% of women with EC present with abnormal bleeding (Clark et al, 2018). We have amended the text to make this clearer: We considered women who did not present with abnormal bleeding to have atypical presentations.
Reviewer 2 Comment 7	Reviewer 2 Response 7
Page 6, line 132: How would this assay distinguish between an endocervical adenocarcinoma and an endometrial adenocarcinoma given the different management strategies? Is this based on IHC alone, HPV in situ hybridization?	The aim of urogenital cytology is to identify women who should be fast tracked for further investigations, including endometrial biopsy +/- hysteroscopy +/- imaging as appropriate. It is not intended to replace standard diagnostics but rather to limit their application to those who are at high risk of cancer. In so doing, healthy women will be reassured quickly, spared the psychological and physical morbidity of invasive testing and costs will be reduced.

	Clearly the management of cervical adenocarcinoma is quite different to the management of EC and it is extremely important to make this distinction. We tested 30 urine and vaginal samples in women with positive urogenital cytology and found those with cervical cancer (n=4) were always high risk HPV positive, whilst those with endometrial cancer were high risk HPV negative (n=26). The residual liquid based cytology sample is sufficient for HPV testing (and this is what has traditionally happened of course in cervical screening – those with positive cytology undergo reflex high risk HPV testing on the residual sample). The presence of koilocytes and dyskaryotic squamous cells are another indicator of HPV-related change in these samples (even in urine).
Reviewer 2 Comment 8	Reviewer 2 Response 8
It is unclear to me what the urine assessment adds to the detection of endometrial cancer. Were any cases detected on urine alone? I understand that urine cytology identified alternate cancers, and do not understand the biologic rationale in those cases, but I am not certain from the data presented that urine cytology adds anything to endometrial cancer detection?	Urine cytology had a lower sensitivity than vaginal cytology for the detection of EC, but its specificity was improved and there were very few false positives. We tested it alongside vaginal sampling to determine whether it might have clinical utility for EC detection, especially as it is so easy to obtain, very acceptable to patients and could revolutionize EC detection in symptomatic and possibly even high-risk asymptomatic women. The discovery that malignant endometrial cells can be detected in urine offers multiple novel opportunities for diagnostics in the future. We therefore feel it is important to report this finding. We have added to the discussion:

	Urine is a particularly attractive biofluid for EC detection because it is straightforward to collect and highly acceptable to patients, offering opportunities for repeat sampling and collection at home.
Reviewer 2 Comment 9	Reviewer 2 Response 9
Page 7, line 140: How was stage 4 cervical cancer not detected on clinical exam? This should not be described as “unexplained” PMB.	Yes, this is a good point. The referring clinician did not examine the patient due to frailty. ‘Unexplained PMB’ describes all patients with PMB who had been referred for urgent diagnostic work up but had not yet received a provisional diagnosis. This means that researchers consenting women with ‘unexplained PMB’ to our study had no idea whether they had sinister underlying pathology, ie they were blind to their eventual cancer outcomes. This strengthens the validity of our approach but a locally advanced cervical tumour would, of course, be diagnosed clinically on examination – our test would not be needed!
Reviewer 2 Comment 10	Reviewer 2 Response 10
Page 7, line 165: What is meant by the phrase, “routine clinical diagnosis...”?	We mean ‘routine clinical diagnostic pathway’. Women either had a normal endometrial thickness on transvaginal ultrasound (<4mm) and were not evaluated further, or had a benign endometrial biopsy. We have changed the sentence to improve clarity: Urine and/or vaginal cytology was positive for a further 12/107 cases where cancer was not diagnosed, giving a false positive rate of 11.2%.
Reviewer 2 Comment 11	Reviewer 2 Response 11
How do the authors suggest providers manage + cytology (with the	As discussed above, positive urogenital cytology would signpost a

methods described in the paper), with negative EMB in clinical practice? With an 11.2% false + rate, which is much higher than endometrial biopsy and D&C, would providers ignore the cytology? Not sure how this would be managed clinically.	woman for further diagnostic testing. If her transvaginal ultrasound scan, hysteroscopy and endometrial biopsy were all normal, she would be reassured that there was no sinister underlying endometrial pathology. Other causes for malignant cytology include bladder cancer (positive urine cytology), cervical cancer (residual cytology sample could be reflex tested for high risk HPV) or ovarian cancer (adnexal mass on transvaginal ultrasound), all of which would be picked up using standard diagnostics.
Reviewer 2 Comment 12	Reviewer 2 Response 12
The authors do not discuss the ability of the vaginal or urine cytology to accurately detect or inform underlying histology. The presence of serous or clear cell carcinoma vs endometrioid histology may have implications in patient counseling, surgical staging and need for preoperative imaging. This needs to be discussed and addressed as part of the manuscript.	The aim of this study was to provide proof of principle that malignant endometrial cells could be identified in minimally invasive urine and vaginal fluid samples. Women with positive cytology would go on to have full invasive diagnostics, including an endometrial biopsy. This is important for determining what treatment should be provided (eg fertility-sparing treatment, full surgical staging including lymphadenectomy, etc). However, it is interesting that the cytologists were sometimes able to distinguish endometrioid from clear cell and serous cancers based on morphology alone. This has not been formally tested, however, and we maintain that a tissue sample is needed for diagnostic and treatment planning purposes in those who test cytology positive.

References

- Clark TJ, Mann CH, Shah N, Khan KS, Song F, Gupta JK. Accuracy of outpatient endometrial biopsy in the diagnosis of endometrial cancer: a systematic quantitative review. *BJOG*. 2002 Mar;109(3):313-21.
- Snijders PJ, Verhoef VM, Arbyn M, Ogilvie G, Minozzi S, Banzi R, van Kemenade FJ, Heideman DA, Meijer CJ. High-risk HPV testing on self-sampled versus clinician-collected specimens: a review on the clinical accuracy and impact on population attendance in cervical cancer screening. *Int J Cancer*. 2013 May 15;132(10):2223-36.
- Sundar S, Balega J, Crosbie E, Drake A, Edmondson R, Fotopoulou C, et al. BGCS uterine cancer guidelines: Recommendations for practice. *Eur J Obstet Gynecol Reprod Biol*. 2017;213:71-97.
- Timmermans A, Opmeer BC, Khan KS, Bachmann LM, Epstein E, Clark TJ, et al. Endometrial thickness measurement for detecting endometrial cancer in women with postmenopausal bleeding: a systematic review and meta-analysis. *Obstetrics and gynecology*. 2010;116(1):160-7.
- van Hanegem N, Prins MM, Bongers MY, Opmeer BC, Sahota DS, Mol BW, Timmermans A. The accuracy of endometrial sampling in women with postmenopausal bleeding: a systematic review and meta-analysis. *Eur J Obstet Gynecol Reprod Biol*. 2016 Feb;197:147-55.
- Visser NCM, Reijnen C, Massuger LFAG, Nagtegaal ID, Bulten J, Pijnenborg JMA. Accuracy of Endometrial Sampling in Endometrial Carcinoma: A Systematic Review and Meta-analysis. *Obstet Gynecol*. 2017 Oct;130(4):803-813.

Reviewers' Comments:

Reviewer #1:

Remarks to the Author:

The authors have provided full and clear replies and have thoroughly revised the manuscript accordingly.

The methods section and discussion have both improved.

Only a few remaining minor comments (and a typo):

Results:

The authors state on page 4, line 74-75 " including 100 with known or suspected EC and 113 with unexplained PMB". Should this read 103 EC?

Discussion:

The discussion has now a clearer style with a more concise conclusion (page 11, "Whilst our data are very promising, the clinical utility of urogenital cytology for EC detection must now be confirmed in a large multicentre prospective study of women with unexplained PMB undergoing investigation.") Could the authors specify the type of trial they envision? What would be the control group? This could be specified in more depth.

Table 1:

On the column depicting all patients, on the "presenting complaints" row, the table shows 192 patients with PMB and 23 with other symptoms, while it seems this should be 202 with PMB and 16 with other symptoms.

Reviewer #2:

Remarks to the Author:

I would like to thank the authors for taking the time to address the comments and recommendations of both reviewers. I find the manuscript acceptable for publication in the current format.

REVIEWERS' COMMENTS:	AUTHOR RESPONSE:
Reviewer #1 (Remarks to the Author): The authors have provided full and clear replies and have thoroughly revised the manuscript accordingly. The methods section and discussion have both improved. Only a few remaining minor comments (and a typo): Results: The authors state on page 4, line 74-75 " including 100 with known or suspected EC and 113 with unexplained PMB". Should this read 103 EC? Discussion: The discussion has now a clearer style with a more concise conclusion (page 11, "Whilst our data are very promising, the clinical utility of urogenital cytology for EC detection must now be confirmed in a large multicentre prospective study of women with unexplained PMB undergoing investigation.") Could the authors specify the type of trial they envision? What would be the control group? This could be specified in more depth. Table 1: On the column depicting all patients, on the "presenting complaints" row, the table shows 192 patients with PMB and 23 with other symptoms, while it seems this should be 202 with PMB and 16 with	Thank you. Thank you. Yes, thank you for pointing this out. We have corrected this typo. Thank you. We have amended this section as follows: Whilst our data are very promising, the clinical utility of urogenital cytology for EC detection must now be confirmed in a large multicentre cross-sectional diagnostic accuracy study of women with unexplained PMB undergoing routine diagnostic investigations, with histology or clinical follow up as the reference standard. Alternative diagnoses should be considered for participants with positive urogenital cytology but negative routine diagnostics. Thank you for pointing out this typo, which we have corrected. Thank you.

other symptoms.

Reviewer #2 (Remarks to the Author):

I would like to thank the authors for taking the time to address the comments and recommendations of both reviewers. I find the manuscript acceptable for publication in the current format.